# Evaluation of the Association of *VDR* rs2228570 Polymorphism with Elite Track and Field Athletes’ Competitive Performance

**DOI:** 10.3390/healthcare11050681

**Published:** 2023-02-25

**Authors:** Celal Bulgay, Işık Bayraktar, Hasan Huseyin Kazan, Damla Selin Yıldırım, Erdal Zorba, Onur Akman, Mehmet Ali Ergun, Mesut Cerit, Korkut Ulucan, Özgür Eken, Halil İbrahim Ceylan, Georgian Badicu, Wilhelm Robert Grosz, Raluca Mijaică

**Affiliations:** 1Sports Science Faculty, Bingol University, Bingol 12000, Turkey; 2Faculty of Sports Sciences, Alanya Alaaddin Keykubat University, Alanya 07450, Turkey; 3Faculty of Medicine, Near East University, 1010-1107 Nicosia, Cyprus; 4Sports Science Faculty, Lokman Hekim University, Ankara 06510, Turkey; 5Sports Science Faculty, Gazi University, Ankara 06560, Turkey; 6Sports Science Faculty, Bayburt University, Bayburt 69000, Turkey; 7Faculty of Medicine, Gazi University, Ankara 06560, Turkey; 8Department of Medical Biology and Genetics, Marmara University, Istanbul 34722, Turkey; 9Department of Physical Education and Sport Teaching, Inonu University, Malatya 44000, Turkey; 10Physical Education and Sports Teaching Department, Kazim Karabekir Faculty of Education, Ataturk University, Erzurum 25240, Turkey; 11Department of Physical Education and Special Motricity, Faculty of Physical Education and Mountain Sports, Transilvania University of Braşov, 500068 Braşov, Romania

**Keywords:** athletics, sprint/power athletes, endurance athletes, *VDR*, rs2228570 polymorphism

## Abstract

The present study aimed to examine the vitamin D receptor (*VDR*), rs2228570 polymorphism, and its effect on elite athletes’ performance. A total of 60 elite athletes (31 sprint/power and 29 endurance) and 20 control/ physically inactive, aged 18–35, voluntarily participated in the study. The International Association of Athletics Federations (IAAF) score scale was used to determine the performance levels of the athletes’ personal best (PB). Whole exome sequencing (WES) was performed by the genomic DNA isolated from the peripheral blood of the participants. Sports type, sex, and competitive performance were chosen as the parameters to compare within and between the groups by linear regression models. The results showed no statistically significant difference between the CC, TC, and TT genotypes within and between the groups (*p* > 0.05). Additionally, our results underlined that there were no statistically significant differences for the association of rs2228570 polymorphism with PBs within the groups of the (*p* > 0.05) athletes. The genetic profile in the selected gene was similar in elite endurance, sprint athletes, and in controls, suggesting that rs2228570 polymorphism does not determine competitive performance in the analyzed athlete cohort.

## 1. Introduction

From the inception of humanity to the present date, the differences created by physical performance and related variables have been regarded as very important parameters in terms of the survival and continuity of generations. Particularly within the last century, in addition to factors that affect physical performance such as lifestyle and environmental-epigenetic interactions, changes in which polymorphisms resulting from genetics also play a role have become the focal point of the scientific world. Since new findings and locations of interaction have been revealed along with their causes as a result of the gradual decoding of human genome codes in the past 25 years, new methods and applications that will contribute to the scientific world within the framework of improving individual effort limits have started to gain momentum as well.

Elite athletes easily adapt to the interaction of genomic and epigenomic features; training applications; and the changes caused in organisms by nutrition, lifestyle, and environmental factors. In addition to genetic adaptation, the reshaping of muscles as a result of training and positive responses to training stimuli are the most apparent reflections of the ability to adapt. When used in athletes who exhibit elite levels of performance, candidate genes that reveal athletic ability serve as important clues that can reveal the reflections of the scope and severity of training applications, load, rest and recovery processes, nutrition, and injury risks on the field or podium [1,2].

In contemporary medicine, a new era titled “personalized medicine” is beginning. The necessity to address a variety of metabolic diseases and support individuals with an approach toward sports activities that are specified based on their genetic infrastructure has been increasingly emphasized in many countries and scientific communities [3]. Within this scope, with the discovery of approximately 250 candidate genes that affect athletic performance up to date, the regulation of methods and applications affecting competitive performance in consideration of the genetic features of individuals and particularly elite athletes, and the implementation of application-oriented exercises within this framework, have started to become increasingly common among trainers and athletes as well [4,5]. In this context, one of the candidate genes that has been considered by researchers in recent years in relation to the improvement of physical effort and athlete health is the vitamin D receptor (*VDR*) gene. The vitamin D receptor (VDR) is a member of the nuclear receptor group. The gene coding VDR, a member of the nuclear hormone receptor gene family, was defined for the first time in 1969. It is localized in 12q13.11 and has a size of 100 kb, and more than 100 polymorphisms were defined [6,7]. Exons 2 and 3 of the gene, which consist of 11 exons, code the DNA binding site, whereas exons 4 and 9 code the ligand binding site.

There are four functionally important polymorphic regions within the *VDR* gene. The most significant variations of these are polymorphisms such as rs1544410 (intron 8), rs2228570 (exon 2), rs731236 (intron 8), and rs7975232 (exon 9). These variations can cause various metabolisms [8,9]. The Fok1 polymorphism (rs2228570) is located at the 5′ end of exon 2 of *VDR* and corresponds to the start codon. As a result of the transition of T to C, ATG–ACG conversion occurs in the first codon and a 424 amino acid variant polypeptide is produced [6,9,10,11,12]. In the absence of this variation, a 427 amino acid polypeptide, which is regarded as normal, is synthesized. It is stated that the VDR protein has effects on parathyroid cells, hematopoietic cells, keratinocytes, pancreatic islet cells, reproductive organs, and the immune system. VDR also exhibits a wide tissue distribution involving endothelial, vascular smooth muscles, and cardiomyocytes [13].

Vitamin D, which carries out its functions through VDR, is a fat-soluble molecule derived from the steroidal hormone family [14]. Vitamin D metabolites adjust calcium homeostasis with the parathyroid hormone D [15]. Vitamin D synthesized in the human body from cholesterol plays a role in mechanisms affecting skeletal (mineral density, stress fractures, etc.) and muscular (resistance and power) structure [16] and is stored in fat cells to be transferred into circulation when needed. Vitamin D plays an important role in the maintenance of mineral balance [17] and the regulation of calcium metabolism and protein synthesis within muscle cells by enabling the absorption of phosphorus and calcium in the kidney and intestines [18]. In the functioning of vitamin D metabolism, it is thought that the functional polymorphisms on the receptor have different levels of impact on individuals [8,18]. Vitamin D plays an important role in the healthy development of muscles and bones and neuromuscular functions [19,20,21]. Vitamin D triggers muscle cells to receive the inorganic phosphate obtained from energy-rich phosphate compounds (ATP and CrP) that play a key role in muscular contraction mechanisms, whereas specific VDRs localized in the cell membrane enable the distribution and regulation of intracellular calcium [22,23].

It is known that an increase in vitamin D levels in an athlete has positive effects on the musculoskeletal system. As a result of increased vitamin D levels, ATP regeneration, protein synthesis [18], and a positive acceleration in the development of physical performance aspects such as explosive muscle strength are observed, as well as a decrease in the risk of stress fractures. On the other hand, in the case of vitamin D deficiency, muscular atrophy linked with different sports branches, a decrease in muscular contraction rate, chronic muscle pain, and prolonged post-training muscle recovery time are observed. In consideration of the role of vitamin D in muscular contraction strength and stimulation processes, it can also have a direct effect on functional conditions [18,24,25]. Recent studies suggest that the 25 (OH) D level in athletes is above 40 ng/mL [26,27,28]. In another similar study, it was found that high serum 25 (OH) D3 levels were directly related to speed, power, vertical jump, and muscle strength [26,28,29]. In long-distance runners whose vitamin D concentrations were found to be below 32 ng/mL, it was observed that the tumor necrosis factor significantly increased following extensive training loads, and low levels of vitamin D as well as anti-inflammatory and pro-inflammatory cytokines that ensure recovery were found.

Vitamin D deficiency is caused by insufficient time and frequency of sunlight exposure [30]. However, in studies conducted in various climates and regions, cases of vitamin D deficiency were reported despite sufficient time and frequency of sunlight exposure. In certain studies, it was stated that even athletes who trained outdoors had low vitamin D levels [31], which could lead to over-training syndrome induced by overload, malnourishment, and insufficient rest in the later stages of training [32].

*VDR* polymorphisms can affect the vitamin D demands of the body at varying levels. Essentially, vitamin D can be acquired through dietary sources, but is predominantly synthesized endogenously from the ultraviolet-B radiation of the sun, and it reinforces muscle mass and strength levels by increasing the hormone levels in the body within the framework of enzymatic processes [33]. In a study investigating the effects of vitamin D intake on athletic performance under different seasonal and climatic conditions, the performances of athletes during the summer and winter seasons were compared, and it was stated that the physical efforts exhibited during the winter season were more efficient [34]. This situation explains the fact that vitamin D deficiency does not only occur through being deprived of solar rays, but genetic and epigenetic interactions can play a key role in vitamin D deficiency as well.

The present study aimed to examine the *VDR* rs2228570 polymorphism and its effects on the performance of elite athletes.

## 2. Methods

### 2.1. Participants

This study involved 60 elite athletes (sprint/power: 11 females (35.5%) and 20 males (64.5%); endurance: 10 females (34.5%) and 19 males (65.5%)) licensed in different clubs and affiliated to the Turkish Athletics Federation. The number of the controls (non-athletes) were 20 (female 6 (30.0%) and male 14 (70.0%)); they were healthy unrelated citizens of Turkey without any competitive sports experience. All the athletes and controls were of Caucasian ancestry. The individuals with any known and declared diseases were excluded from this study. The athletes were categorized as either sprint/power or endurance athletes, as determined by the distance, duration, and energy requirements of their events. All athletes were ranked in the top 10 in their sport disciplined nationally. Those in the elite group had participated in international competitions such as the Olympic Games, European Championships, Universiade, Mediterranean Games, and Balkan Championship. The sprint/power group included sprint and power athletes whose events demand predominantly anaerobic energy production. The athletes in this group were 100–400 m runners, jumpers, and throwers. The endurance athlete group included athletes competing in long-distance events demanding predominantly aerobic energy production. This group included 3000 m, 5000 m, 10,000 m, and marathon runners. The athletes in this group (n = 31) were 100–400 m 123 runners (n = 9), jumpers (n = 3), and throwers (n = 19). The endurance athlete group (n = 29) included athletes competing in long-distance events demanding predominantly aerobic energy production. This group included 3000 m (n = 12); 5000 m (n = 5); 10,000 m (n = 4); and marathon (n = 8) runners.

### 2.2. Study Design

This study was carried out in accordance with the Declaration of Helsinki and approval was obtained from the Gazi University Non-Interventional Clinical Research Ethics Committee with the decision dated April 05, 2021 and numbered 09. The informed voluntary consent and demographic information forms were applied for the athletes and control groups before the measurements.

### 2.3. Personal Best (PB)

The International Association of Athletics Federations (IAAF; World Athletics) score scale was used to determine the performance levels of the athletes depending on their personal best/competitive performance [35]. For instance, the IAAF score scale of a male athlete who runs 100 m in 10.05 sec is 1189, whereas that of a marathon runner who completes the race in 2 h 20 min 11 sec is 997. Thus, the performance scale of the marathon runner is less than that of the 100 m runner. The IAAF scales are useful for the determination of performances of athletes from diverse athletics events and genders.

### 2.4. Whole Exome Sequencing

Total genomic DNA was isolated from the peripheral venous blood of the participants (4cc) for further genetic screenings using a DNeasy Blood and Tissue Kit (Qiagen, Germany) according to the supplier’s instructions. The quality of the isolated DNA was verified using 1% agarose gel electrophoresis and NanoDrop (NanoDrop 1000 Spectrophotometer; Thermo Scientific, USA) according to optical density ratios, and the concentration was determined by NanoDrop.

Whole exome sequencing (WES) was performed after library preparation by Twist Human Comprehensive Exome Panel (Twist Biosciences, USA) according to the instructions of the supplier. Briefly, DNA was fragmented enzymatically, size selection was carried out, and hybridization was applied using Twist Hybridization probes and Dynabeads™ MyOne™ Streptavidin T1 (Invitrogen, USA), and the library was enriched by polymerase chain reaction (PCR). The concentration and size of the libraries were determined, and sequences were performed using Illumina NextSeq500 according to the manufacturer’s standard protocol.

Raw data were processed to by the genome analysis toolkit (GATK) [36]. The HaplotypeCaller program was used to obtain binary alignment map (BAM) files and subsequently produce an output variant call format (VCF) file via the GRCh38/hg38 reference genome. Variants were annotated by ANNOVAR [37] and each single nucleotide polymorphism (SNPs) was analyzed manually. All molecular analyzes were performed in Gazi University Medical Genetics Laboratory.

### 2.5. Statistical Analyses

The SPSS statistical package version 25.0 for Mac was used to perform all statistical analyses. In the evaluation of the data, descriptive statistical methods (number, percentage, and mean) were used. Before performing any analysis on the data, the study determined whether they met the requirements for parametric tests. To that end, the variables were tested for normality, whereas Kolmogorov–Smirnov and Shapiro–Wilk (*p* = 0.200; 0.785, respectively) tests were used for homogeneity of variance. As result of these tests, parametric tests were performed for the variables distributed. Genotype and allele frequencies were calculated for the polymorphism, and the Hardy–Weinberg equilibrium (HWE) was assessed using the chi-squared (χ^2^) test. Allele and genotype frequencies and any associations between the polymorphism and the athletic parameters were assessed by SNPStats (https://www.snpstats.net/start.htm, acceded on 11 October 2022) [38] using linear regression models, which were used to assess the score of variation in PB with linear regression multiple inheritance models: co-dominant, dominant, recessive, over-dominant, and additive. To confirm the results obtained using the linear regression models, we also analyzed the data by means of the one-way analysis of covariance (ANCOVA), adjusting for sex and sports experience. Data were significant when *p* < 0.05.

## 3. Results

The present study aims to decipher any possible association of the rs2228570 polymorphism with the competitive performances of a group of elite athletes (mean age ± SD: 25.1 ± 4.8; height (cm): 174.97 ± 7.9; body weight (kg) 72.5 ± 22.4; sports experience (year) 9.4 ± 4.8; personal-best (PB) = 1005.63 ± 94.55) in the presence of a control group (mean age ± SD: 23.5 ± 7.1). Three groups (speed/throw/jump) and long-distance athletes and controls have been chosen to assess this aim.

The genotype and allele frequencies were determined. According to the results, there were not any statistically significant different between the CC, TC, and TT genotypes within and between the groups (Table 1; *p* > 0.05). For allele frequencies, although the number of T alleles (n = 112) was higher than the C allele (n = 48), it was not statistically significant (*p* > 0.05; Table 1).

Additionally, when we compare the genotype or allele distribution for both sex and sports experience, no significant difference was detected (data not shown). Therefore, to increase the power of the statistical results, we decided to pool the sample.

*VDR* rs2228570 polymorphism was also evaluated to determine whether it was associated with personal bests (PBs) using different genetic models, codominant, dominant, recessive, and over-dominant. Our results underlined that there was not any significance for the association of rs2228570 polymorphism with PBs within the athlete group (*p* > 0.05; Table 2).

## 4. Discussion

A predisposition towards sports activities in humans is a large variable factor. Although muscle mass and function were initially listed among the main reasons behind this variability, it has been stated within the framework of the scientific data reported in recent years that elite athletic performance ability originates from genetics. In this context, it is thought that high levels of effort can be achieved through the regulation of variables such as the type of nutrition, as well as the severity and duration of training programs planned in line with the properties of the related candidate genes in addition to the existing genetic infrastructure of the individual. In recent years, genetic tests and applications that reveal branch-specific athletic ability have been among the most frequently emphasized concepts, particularly in the field of the sports industry. In tests carried out within this scope, the presence of candidate genes that reveal capability is regarded as indicating the capability of individuals, and outstanding candidate genes are taken into consideration, particularly in elite athletes. Genetic differences allow certain individuals to reach their goals in a very short period of time, whereas others can achieve high performance following a very long period. Although there are various body types among humans, body type and physiological characteristics depend on the distribution in the genetic structure, which varies based on environmental and epigenetic factors. These differences determine the outcome for elite athletes [39].

Similarly, the type and efficiency of nutrition are also affected by genetic factors. For example, the microbiome is heavily affected by host genetics [40]. Polymorphisms in various genes, including *VDR* rs2228570, can differentiate individual responses to training loads by influencing muscle strength and resistance [41]. VDR can affect the vitamin D needs of an individual. Within this framework, differences between individuals that result from genetics, common environment, and epigenetic factors determine the functionality of primary nutritional sources and supplements within the metabolism. Similarly, vitamin D absorption levels originating from *VDR* gene polymorphisms differ among individuals. Therefore, functional polymorphisms on the receptor play a very significant role in terms of understanding the effects of vitamin D metabolites on athletic performance and a healthy lifestyle [42].

Vitamin D is among the bioactive molecules required for bone mineral density, as well as muscular contraction and regeneration. Additionally, it accelerates regeneration time by reducing the level of inflammation that occurs after exhaustive training loads. Within this framework, it was stated in previous studies that the daily level of vitamin D required for athletes should be at high values such as approximately 32–40 ng/mL (20 ng/mL for sedentary individuals). However, variables originating from individual differences (climate; geography; genetic structure; type, severity, and duration of the training; etc.) can affect levels of vitamin D use. Though sufficient levels of vitamin D intake allow for the maintenance and development of athletic performance, the reduction of said vitamin levels inhibits muscle relaxation required after training and increases muscle pain while also triggering injury risk and stress fractures by causing muscle power loss and reducing bone mineral density [21].

In the findings of a study comparing elite athletes who were found to have insufficient vitamin D levels (n = 61) and the control group (n = 30) (eight weeks/daily vitamin D intake of 5000 IU), it was found that 25(OH)D levels observed in serum improved exercise performance (sprint, vertical jump) (Close et al., 2013), whereas another similar study reported that serum 25(OH) D concentrations measured on dancers (n = 98) were low (73%) due to vitamin D deficiency [43]. Additionally, in another study conducted on lab rats, it was observed that vitamin D reduced injury risk in anaerobic training loads by increasing the volume and number of type II muscle fibers, making positive contributions to maximal sprint and power capabilities [44].

In another study conducted by Allison et al. [45] (n = 506), vitamin D levels and structural coronary parameters were compared within the framework of the effects of vitamin D on the cardiovascular system and it was found that the structural parameters of the athletes below the 25(OH)D levels observed in serum were lower compared to the others [45].

*VDR* is among the first polymorphisms studied in the determination of the link to athletic performance. In the studies carried out within this framework, it was observed that the Fok1 polymorphism had a greater impact on bone mineral density along with exercise and nutrition in individuals with the CC homozygous genotype [46], and increased the risk of sarcopenia [9,47]. In addition to the impact of the related polymorphism on changes in bone mineral density linked with endurance training, it was reported to be related to environmental factors such as calcium intake [48,49]. Polymorphisms in different genes, including *VDR*, can impact muscle strength and alter responses to training stimuli [33].

Certain previous studies reported that the rs2228570 polymorphism increased the risk of stress fractures in athletes performing heavy exercise and that it is necessary to analyze the said genetic variable to create healthy training programs [50]. In a similar study conducted with young soldiers consisting of males and females (n = 385), 268 single nucleotide polymorphisms (SNP), including the *VDR* gene and 17 gene regions, were analyzed, and it was stated that the *VDR* gene is an important determinant in the onset of stress fractures [51]. In the findings of another similar study conducted on military personnel, it was found that patients with stress fractures were much more likely to possess the rs2228570 polymorphism [52]. In a previous study investigating the impact of VDR gene polymorphisms on the longitudinal changes in hormones related to bone mass, bones, and calcium in adolescent football players (n = 46; 11.8–14.2 years old), it was stated that the *VDR* gene polymorphism affected bone mass in adolescent football players (41.3% on the CC genotype rate, 47.3% on the CT genotype rate, and 10.9% on the TT genotype rate) and that its impact on bone mineralization likely occurred in early stages of puberty during bone maturation [53]. As in chronic obstructive pulmonary disease (COPD) patients, it was observed that individuals with systemic diseases had weaker quadriceps muscles in the CC alleles compared to the CT and TT genotypes [54].

Similarly, the study conducted by Eken et al. [6] stated that CC genotypes and C alleles were more dominant in elite athletes [6]. In contrast with the said study, in the present study, although the frequency of the T allele (n = 48) was higher compared to the C allele among all participants, no statistically significant intragroup and intergroup difference was observed (Table 1; *p* > 0.05). By contrast, another study (n = 206 men and women; 50–81 years old) reported that the Fok1 polymorphism had a greater effect on bone mineral density in lengthy and low-intensity aerobic and strength training applications, although the said interaction was unrelated to the development of aerobic resistance training levels [55]. Supporting the findings of the present study, in the study conducted by Morucci et al. [56] on gymnasts (n = 80), it was stated that there was no relationship between *VDR* genotypes and athletic performance in terms of genotype [56]. Similarly, in another study conducted by Gavin et al. [57] on female athletes (n = 62), the relationship between *VDR* and muscle strength was examined; however, no relation was found [57].

In the vertical jump performance tests carried out by Massidda et al. [58] on professional male football players (n = 90), the percentage of participants with the CC genotype (46.3%) was found to be higher compared to the CT (38.8%) and TT (14.8%) genotypes [58]. Differently from the results of the present study, in another study similar to that of Micheli et al. [59], the rate of young male football players (n = 125) with the CC genotype was found as 52%, whereas this percentage was 34% for the CT genotype and 14% for the TT genotype, stating that these genotype results can serve as important indicators in terms of the predisposition towards football [59].

The fact that the rate of football players with the CC genotype was found to be higher in both of the aforementioned studies compared to the competitive performance findings in the present study can be interpreted as notable differences between sports disciplines in terms of performance level, individual features, and the rate of use of energy sources. Though both studies show similarities with the analyses performed in the present study, this is not adequate to explain the reasons why the frequencies of the C or T alleles were high or low. The present study found no significant intragroup or intergroup difference (speed/throw/jump, long distance, and control) between the CC, TC, and TT genotypes. This was also the case in genotype or allele distributions within the framework of both genders and sports experiences, and in the light of the findings obtained within this scope, no correlation was found between the *VDR* rs2228570 polymorphism and the personal bests (PBs) in the athlete groups (Table 2; *p* > 0.05). The fact that the number of studies and data on the cellular effects of this polymorphism are insufficient may lead to the conclusion that the T or C alleles provide a predisposition toward the status of elite athletes. However, the impact of either allele on a molecular level is not clearly known, which will cause their contribution or ineffectiveness toward individual performance to remain unclear.

With approximately three billion nucleotides in the human genome, the number of nucleotide combinations that can affect gene activity is essentially infinite. Beyond the aforementioned studies, many studies show that genes play an important role in the determination of athletic performance. Considering the improvements achieved in the past 20 years, it can be foreseen that the genetic elements affecting athletic performance phenotypes and training stimuli will become more comprehensible in the following years with multigene and multifactorial approaches. On the other hand, based on the available data, it is certain that the effects of the candidate genes involved in athletic performance development as the desired phenotype are not very clear, and that further studies are needed.

Although the present study underlined the insignificance between the related polymorphism and athletic parameters, there were some limitations that could affect the results. The single gene and polymorphism approach may alter the results of the associations in particular pathways where more than one protein is involved. Thus, a cumulative evaluation study for all genes and SNPs is under progress by our research group. In our study, biochemical measurements such as the athletes’ vitamin D levels were not measured, and it was not checked whether the athletes took vitamin D before the study. In future studies, more comprehensive studies can be conducted by taking these factors into consideration. Nevertheless, such approaches are planned as further studies. Furthermore, the number of the participants was relatively lower, which could influence the results, but this criterion was restricted to the number of elite athletes in the population.

## 5. Conclusions

The unclarity of the *VDR* Fok1 rs2228570 polymorphism on a molecular level, as well as the limited number of studies on athletic performance in the literature, stand out as limiting factors in the interpretation of the outcomes of the present study. Within this framework, it is thought that the present study will contribute to the developments in the field of sports sciences along with the limited number of other similar studies.

In conclusion, the genetic profile in the selected gene was similar in elite endurance, sprint athletes, and in controls, suggesting that rs2228570 polymorphism does not determine competitive performance in the analyzed athlete cohort. Additionally, it is foreseen that conducting similar studies in the future with athlete groups in different sports branches will allow for the effects of the *VDR* gene on athletic performance to be identified more clearly.

## Figures and Tables

**Table 1 healthcare-11-00681-t001:** Genotype and allele frequencies of *VDR* rs2228570 polymorphism in elite athletes and controls.

	Genotype			*p*-Value	Allele		*p*-Value
	CC	TC	TT	0.717	C	T	0.618
Sprint/Power	1(3.2%)	17(54.8%)	13(42.0%)		19(30.6%)	43(69.4%)	
Endurance	1(3.4%)	13(44.8%)	15(51.7%)		15(25.9%)	43(74.1%)	
Controls	2(10.0%)	10(50.0%)	8(40.0%)		14(29.6%)	26(70.4%)	

Statistically significant differences (*p* < 0.05); T: thymine, C: cytosine.

**Table 2 healthcare-11-00681-t002:** Association analysis of the *VDR* rs2228570 polymorphism with competitive performance.

Model	Genotype	n	Mean Score (PB)	Difference (95% CI)	*p*-Value
Co-dominant	CC	28	1011	0.00	0.72
	TC	30	997	−18 (−65 to 28)	
	TT	2	1009	8 (−122 to 138)	
Dominant	CC	28	1011	0.00	0.48
	TC-TT	32	997	−17 (−62 to 28)	
Recessive	CC-TC	58	1003	0.00	0.79
	TT	2	1009	17 (−110 to 145)	
Over-dominant	CC-TT	30	1010	0.00	0.42
	TC	30	997	−19(−64 to 27)	

Statistically significant differences (*p* < 0.05); adjusted by sports experience + sex.

## Data Availability

The data presented in this study are available on request from the corresponding authors.

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
