# Peer review of "Evaluation of the Association of *VDR* rs2228570 Polymorphism with Elite Track and Field Athletes’ Competitive Performance"

_healthcare, 2023, doi:10.3390/healthcare11050681_

Round 1

Reviewer 1 Report

Hello.

The article falls into both the field and addresses an interesting topic.

The research presentation is well done, orderly and explicit.

The argument and the introduction are solid and support the idea of research. The methods are clearly presented, as well as the content of the research.

The main question would be what future research directions does this research open up? Can the methodology be applied in other sports disciplines as well?

The area in which the article falls is interesting, in the course of research and as it is highlighted in the conclusions, the limitations make the advance slow.

I cannot analyze from a technical point of view to propose additional controls.

The presented conclusions are supported by the content of the research, and the references are appropriate to the topic addressed.

I have no comments on the tables and figures.

Congratulations. All the best!.

Author Response

Dear Reviewer, 

Please see the paper. 

Regards, 

Reviewer 2 Report

The manuscript titled Evaluation of the Association of VDR polymorphism rs2228570 with Elite Track and Field Athletes’ Competitive Performance adds further knowldge on the association between genetics and sport performance, focusing the attension on VDR gene.

The manuscript, in my opinion, can be accepted for publication after minor revision, in details:

1) Introduction: It is not clear why authors chose to analyse only one SNP (since they have all the genome available), and why just the rs2228570

2)   Methods: Please add more details for the sampling (criterion of inclusion and exclusion), particularly the origin of the sample. Authores declare the athletes are affiliated to Turkish federation and the controls are Turkish citizens but no “real” origin are defined, and it is important to have an omogeneus sample.  

3)     Methods: have the authors administer a questionnaire to participants? A declaration of no Vitamin D intake should be required.

4)     Methods: Control sample should be incremented.

5)      Methods: it is not clear why authors prefere the use of electrophoresis to verifiy the quality of the samples rather than the use of nanodrop (ratio 260/280, and 260/230)

6)      Results:     Please, can add some words about Personal best for readers not expert in sport (e.g. how do uniform the different PB of all the athletes?)

7)      Discussion: page 7, line 326, the reference 59 is not Massidda et al.

8)      Discussion:  page 8 line 343: I would change physical activity into status of elite athletes, since with PB you evaluate the status of elite not the motor ability.

Author Response

(The authors gave the same response as above.)
